# The influence of the native lung on early outcomes and survival after single lung transplantation

Francisco Javier Gonzalez, Enriqueta Alvarez, Paula Moreno, David Poveda, Eloisa Ruiz◉, Alba Maria Fernandez, Angel Salvatierra, Antonio Alvarez◉*

Thoracic Surgery and Lung Transplantation Unit, University Hospital Reina Sofia, Cordoba, Spain

* aalvarez53@gmail.com

## Abstract

### Objective

To determine whether problems arising in the native lung may influence the short-term outcomes and survival after single lung transplantation (SLT), and therefore should be taken into consideration when selecting the transplant procedure.

### Patients and methods

Retrospective review of 258 lung transplants performed between June 2012 and June 2019. Among them, 161 SLT were selected for the analysis. Complications in the native lung were recorded and distributed into two groups: early and late complications (within 30 days or after 30 days post-transplant). Donor and recipient preoperative factors, 30-day mortality and survival were analysed and compared between groups by univariable and multivariable analyses, and adjusting for transplant indication.

### Results

There were 161 patients (126M/35F; 57±7 years) transplanted for emphysema (COPD) (n = 72), pulmonary fibrosis (IPF) (n = 77), or other indications (n = 12). Forty-nine patients (30%) presented complications in the native lung. Thirty-day mortality did not differ between patients with or without early complications (6% vs. 12% respectively; p = 0.56). Twelve patients died due to a native lung complication (7.4% of patients; 24% of all deaths). Survival (1,3,5 years) without vs. with late complications: COPD (89%, 86%, 80% vs. 86%, 71%, 51%; p = 0.04); IPF (83%, 77%, 72% vs. 93%, 68%, 58%; p = 0.65). Among 30-day survivors: COPD (94%, 91%, 84% vs. 86%, 71%, 51%; p = 0.01); IPF (93%, 86%, 81% vs. 93%, 68%, 58%; p = 0.19). Native lung complications were associated to longer ICU stay (10±17 vs. 33±96 days; p<0.001), longer postoperative intubation (41±85 vs. 99±318 hours; p = 0.006), and longer hospital stay (30±24 vs. 45±34 days; p = 0.03). The presence of late native lung problems predicted survival in COPD patients (OR: 2.55; p = 0.07).

**Data Availability Statement:** All relevant data are within the paper and its Supporting Information files.

**Funding:** The authors received no specific funding for this work.

**Competing interests:** The authors have declared that no competing interests exist.

## Conclusion

The native lung is a source of morbidity in the short-term and mortality in the long-term after lung transplantation. This should be taken into consideration when choosing the transplant procedure, especially in COPD patients.

## Introduction

Lung transplantation is a well established, and lifesaving treatment for end-stage pulmonary diseases, especially in whom all medical treatment has failed. After more than 40 years of worldwide clinical experience, both single (SLT) and bilateral lung transplants (BLT) are widely accepted, and its selection rely on clinical features, such as age, pulmonary arterial pressure or infectious status [1].

In the first lung transplant era, SLT was the procedure of choice. Nonetheless, in the last decade, this trend has been inverted, and BLT is the most frequently performed after demonstrating better survival, among other several advantages in comparison with SLT [2, 3].

Despite all this, the topic remains controversial, and some authors still recommend performing SLT in older patients with specific diseases such as pulmonary fibrosis [4], especially given the limited organ availability and the unacceptable mortality rate while on the waiting list [5].

The longer survival of BLT has been related to a better postoperative lung function and less chronic lung allograft dysfunction (CLAD) compared with unilateral procedures [6, 7]. However, there is little evidence demonstrating that the native lung could be the cause of a worst survival after a SLT.

Therefore, the aim of this study is to determine whether the native lung after SLT could be the source of morbidity and mortality that might explain, at least in part, the longer survival of BLT.

## Patients and methods

### Study design

This is an observational analytic retrospective case-control study to determine the influence of native lung complications on early outcomes and survival after SLT for IPF or COPD. The primary end-points were 30-day mortality and survival. For this purpose, the medical records of 258 patients transplanted between June 2012 and June 2019 at our Institution were retrospectively reviewed from the pulmonary transplantation database. Among them, 161 SLT were selected for the analysis. Complications in the native lung were recorded and distributed into two groups: early and late complications (within 30 days or after 30 days post-transplant).

### Lung transplantation procedure

Lungs were retrieved from brain death donors using our standard protocol of cardiopulmonary harvesting [8]. Either a right or left SLT was performed in all cases through a standard posterolateral thoracotomy or through an anterior thoracotomy. The surgical procedure and the postoperative standard of care for lung transplant recipients were performed as previously reported by our group [9].

## Data collection

Recipient preoperative data included: age, gender, steroid therapy, preoperative ventilation, native lung perfusion, indication for lung transplantation (COPD, IPF, or other indication) and comorbidities (airway colonizations, bullae, granulomas, lung hyperinflation, bronchiectasis).

Surgical and early postoperative data included: side of lung transplant, use of intraoperative extracorporeal life support (ECLS) (cardiopulmonary bypass or extracorporeal membrane oxygenation—ECMO), ischemic time, duration of mechanical ventilation, ICU stay, need of intraoperative or postoperative lung volume reduction surgery, and 30-day mortality.

Late postoperative data included: late native complications, overall mortality, and survival.

Data were analysed and compared between groups (early and late complications) by univariable and multivariable analyses, adjusting for transplant indication (COPD or IPF).

## Definitions

Recipients were considered on preoperative steroid therapy when receiving steroids at doses above 15mg/kg for 30 days just before transplantation.

## Statistical analysis

**Univariable analysis.** We compared complicated vs. non-complicated recipients by either Pearson's $\chi^2$ or Fisher's exact test for categorical variables, and either unpaired t-test or Mann-Whitney U-test for quantitative variables. These comparisons were performed in the overall series, and in each subgroup of patients transplanted either for COPD or IPF.

Given the large sample size of the study population, we assumed homogeneous variances and normal distribution in most of the analyses. We used parametric tests when more than 30 cases for each group were compared and non-parametric tests when less than 30 cases for each group were compared. Pre-tests for normality were not performed.

Survival was analysed and compared using the Kaplan–Meier method and log-rank test.

**Multivariable analysis.** To determine independent predictors of mortality, those variables exhibiting $p$ values below 0.1 in the univariable analyses entered into a multivariable Cox-regression analysis (forward stepwise likelihood ratio). Those variables with $p$ values below 0.05 in the final model were judged to be independent predictors of mortality.

Continuous variables are expressed as means ± standard deviation. Categorical variables are expressed as counts and proportions with 95% confidence intervals (95% CI). Differences with $p$ values <0.05 were considered significant. The statistical analysis was performed using SPSS (SPSS 20.0 for Mac: SPSS, Inc., Chicago, IL, USA).

The data set from which the analysis was performed is available in S1 File. This file contains fully anonymized data before accession to the analysis, and includes the date range (month and year) during which patients' medical records were accessed. The data file belongs to the Lung Transplant Unit, Hospital Universitario Reina Sofía, Cordoba, Spain.

## Ethical statement

The present study followed the World Medical Association Declaration of Helsinki-Ethical Principles for Medical Research involving human subjects. All patients had signed the Informed Consent at the time of inclusion in waiting list to undergo the procedure and to use their medical records for research.

Our Institutional Review Board approved this study and waived the need for additional specific informed consent for the purposes of the present study.

None of the transplant donors was from a vulnerable population and all donors or next of kin provided written informed consent that was freely given. A blank example of the form used to obtain consent from donors is included (S2 File).

Presumed consent was introduced in Spain by law in 1979. The law establishes that absence of explicit refusal automatically makes the patient a potential donor, but requires that a patient's possible refusal to donate should be sought by checking their belongings and consulting proxy decision makers. Since most patients have not registered as donors and do not carry donor cards, Spanish transplant coordinators usually have to establish the patient's wishes through discussion with the family. In practice, organ procurement is not undertaken if the family refuses the donation. Therefore, even though we have a presumed consent policy, we do not apply it in practice. We always approach the relatives, explaining to them the patient's health conditions and we try to find out whether the individual wanted to be an organ donor or not. If the relatives oppose to deceased organ donation, we do not go on with it.

The Spanish Real Decreto 2070–1999 forbids any person from obtaining any kind of financial compensation for human organs and frames organ donation as a voluntary and altruistic act. Therefore, obtaining reimbursements or financial compensations for organ donation in Spain is illegal [10].

## Results

### Overall patient series

From June 2012 to June 2019, 258 lung transplants were performed at our Institution. Among them, 161 were single lung transplants (study group): 126 (78%) males, 35 (22%) females. The mean age was 57±7 [33–68] years old.

Indications for SLT were COPD in 72 patients, IPF in 77 patients, and other indications in 12 cases.

The overall rate of native complications was 30% (49 patients). Fifteen patients developed early complications (9%), and 41 patients developed late complications (25%).

### Native lung complications analysis

Complications in the native lung within 30 days post-transplant (early complications) included atelectasis requiring bronchoscopic toilette of bronchial secretions, pneumothorax requiring pleural drainage, pneumonia, pleural effusion and air leaks following lung volume reduction surgery in the native lung following SLT for COPD (Table 1).

**Table 1. Summary of complications in the native lung in patients after single lung transplantation.**

| Complication | Early | Late |
|---|---|---|
|  | *n(%)* | *n(%)* |
| **Atelectasis** | 3 (2) | 4 (2) |
| **Pneumothorax** | 4 (2.5) | 2 (1) |
| **Pneumonia** | 2 (1) | 8 (5) |
| **Pleural effusion** | 3 (2) |  |
| **Air leaks**[a] | 3 (2) |  |
| **Hyperinflation** |  | 15 (9) |
| **Carcinoma** |  | 11 (7) |
| **Fibrosis progression** |  | 1 (0.6) |

[a]After lung volume reduction surgery in the native lung.

Late complications were more frequently related to progression of COPD changes in the native lung (lung hyperinflation leading to mediastinal shift and subsequent dysfunction of the implanted lung, secondary to parenchymal compression), and to the development of neoplasms (up to 11 COPD patients developed bronchogenic carcinoma in the native lung) (Table 1).

## Mortality

Overall mortality was 22 (30%) for COPD and 27 (35%) for IPF patients. 30-day mortality was higher for IPF patients than for COPD patients: 15 (19%) vs. 6 (8%) respectively (p = 0.04). Interestingly, patients without native lung complications presented higher 30-day mortality than those with some native lung complication: 16 (15%) vs. 1 (2%) respectively (p = 0.008). These differences persisted, but not significantly, when analysing 30-day mortality by transplant indication (Fig 1).

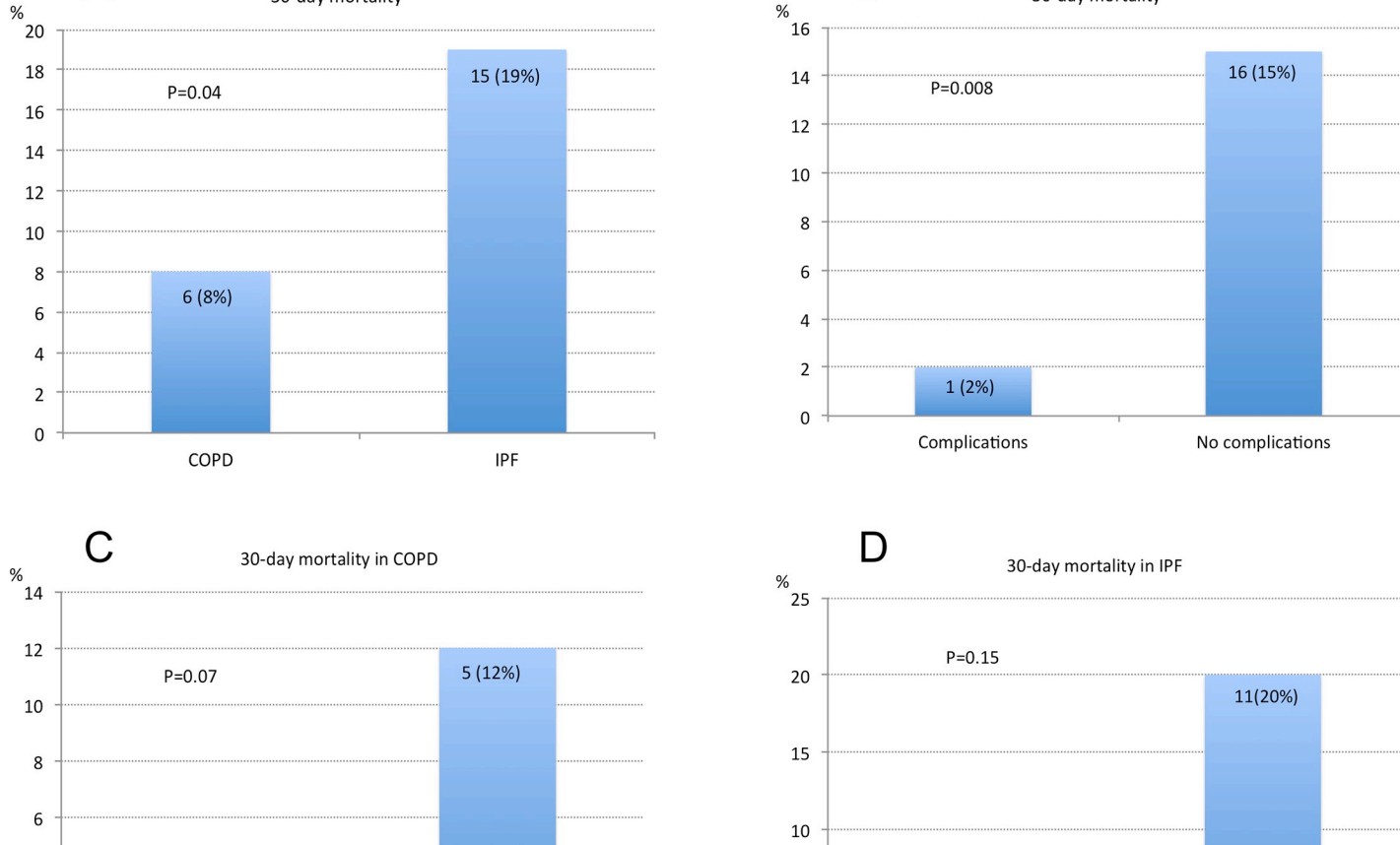

**Fig 1. 30-day mortality.** A) 30-day mortality of COPD vs. IPF patients after single lung transplantation. B) 30-day mortality in patients with or without native lung complications (overall study group). C) 30-day mortality in COPD patients with or without native lung complications. D) 30-day mortality in IPF patients with or without native lung complications.

Among 22 deaths in COPD recipients, 5 were related to native lung complications (23% of COPD deaths). Among 27 deaths in IPF patients, 7 were related to native lung complications (26% of IPF deaths). Overall, 12 deaths were due to native lung complications (7.4% of all patients, and 24,4% of all deaths). Native-related causes of death were lung cancer in 5 COPD patients, pneumonia in 6 IPF patients and fibrosis progression in another IPF patient.

## Comparative analysis between recipients with and without native lung complications

Patients with native lung complications required prolonged postoperative ventilation, and longer ICU and hospital stay. Some preoperative conditions were associated to native lung complications, such as the presence of airway colonizations, bullae, lung hyperinflation and the need of volume reduction surgery (Table 2). Interestingly, 30-day mortality was more frequent in patients without native lung complications, indicating that the causes of early mortality post-transplantation are unrelated to the native lung (Table 2).

## Survival

Survival was not influenced by the presence of native lung complications in IPF patients, neither overall nor when comparing early vs. late complications (Fig 2). On the contrary, those

**Table 2. Comparative data between patients with and without native lung complications following single lung transplantation (overall group).**

| NATIVE LUNG COMPLICATIONS | NO (n = 112) | | YES (n = 49) | | p |
|---|---|---|---|---|---|
| | | *95% CI* | | *95% CI* | |
| Recipient age (years) | 57±7 | | 58±7 | | 0.86 |
| Recipient gender: | | | | | 0.48 |
| Male | 80 (71) | | 38 (77) | | |
| Female | 32 (29) | | 11 (23) | | |
| Preop. Steroids | 20 (19) | 12–26 | 14 (28) | 16–40 | 0.14 |
| Ischemic time (min) | 307±57 | 297–317 | 295±53 | 281–309 | 0.57 |
| CPB / ECMO | 14 (12) | 6–18 | 2 (4) | 0–8 | 0.05 |
| ICU stay (days) | 10±17 | 7–13 | 33±96 | 7–59 | <0.001 |
| Postop. Intubation (h) | 41±85 | 26–56 | 99±318 | 54–144 | 0.006 |
| Hospital stay (days) | 30±24 | 26–34 | 45±34 | 36–54 | 0.03 |
| PaO$_2$/FiO$_2$ 24h postop. (mm Hg) | 345±146 | 318–372 | 316±182 | 265–367 | 0.35 |
| Comorbidities | 39 (35) | 26–44 | 24 (49) | 35–63 | 0.04 |
| Native lung perfusion (%) | 52±9 | 51–53 | 48±13 | 45–51 | 0.29 |
| Native lung side (left/right) | 30/82 (26/74) | 18-34/66-82 | 10/39 (20/80) | 9-31/69-91 | 0.29 |
| CLAD | 4 (3) | 0–6 | 5 (10) | 2–18 | 0.12 |
| *Pre-transplant status* | | | | | |
| Airway colonizations | 4 (3) | 0–6 | 15 (30) | 18–42 | <0.001 |
| Bullae | 36 (32) | 24–40 | 30 (61) | 48–74 | 0.003 |
| Granulomas | 7 (6) | 2–10 | 7 (14) | 5–23 | 0.13 |
| Hyperinflation | 0 | | 3 (6) | 0–12 | 0.03 |
| Mechanical ventilation | 2 (2) | 0–4 | 0 | | 0.44 |
| LVRS | 0 | | 3 (6) | 0–12 | 0.03 |
| 30-day mortality | 16 (14) | 8–20 | 1 (2) | 0–4 | 0.008 |

CPB: cardiopulmonary bypass; CLAD: chronic lung allograft dysfunction; ECMO: extracorporeal membrane oxygenation; LVRS: lung volume reduction surgery.

Quantitative variables are expressed as mean ± standard deviation. Qualitative variables are expressed as counts and proportions within each column, in parenthesis.

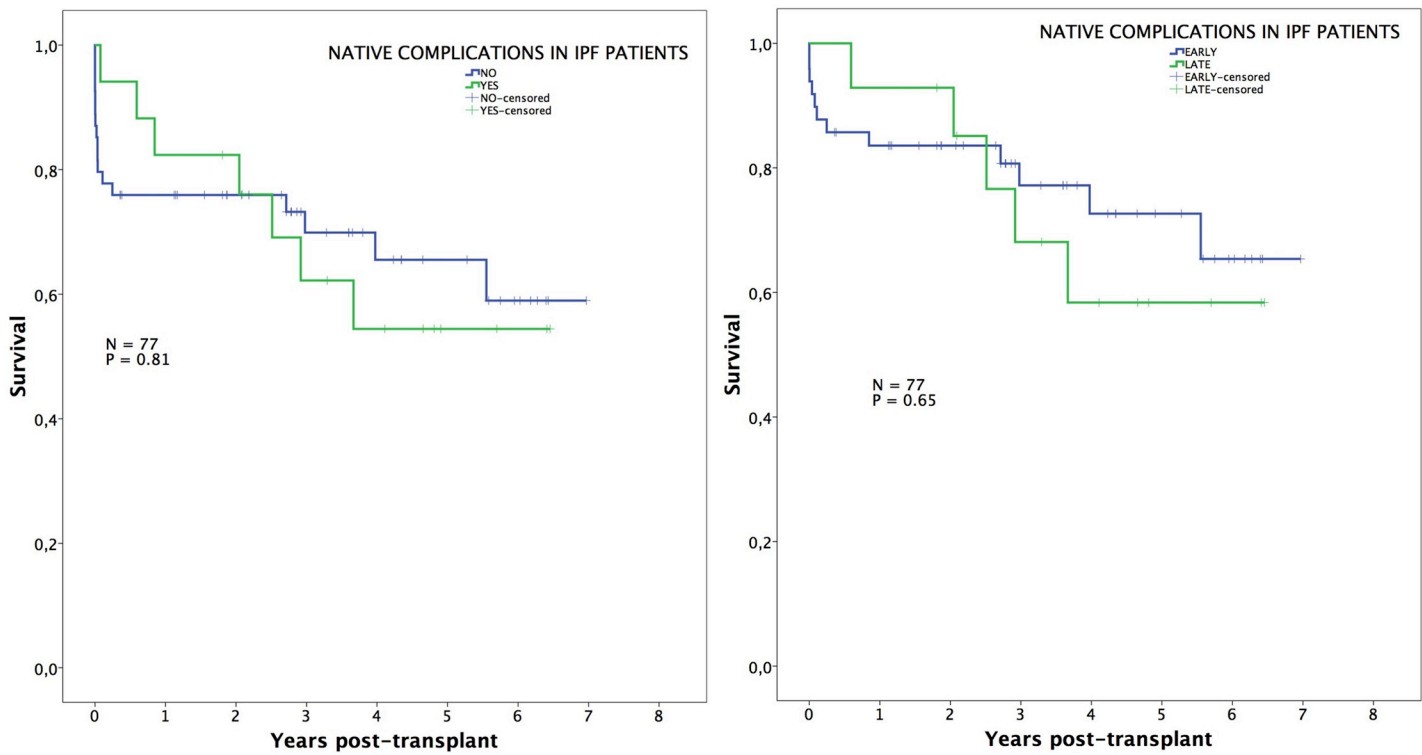

**Fig 2. Survival in IPF patients.** Post-transplant survival of IPF patients comparing those with or without native lung complications (left), and those with early or late native lung complications (right).

patients transplanted for COPD exhibited worse survival when native lung complications arose. This is especially true when comparing survival between those COPD patients with early complications vs. late complications (89%, 86%, 80% vs. 86%, 71%, 51%; p = 0.04) (Fig 3).

In an additional analysis comparing survival among those patients surviving beyond 30 days post-transplant, we observed that long-term survival was significantly impaired in COPD patients developing late complications (neoplasms and lung hyperinflation) (94%, 91%, 84% vs. 86%, 71%, 51%; p = 0.01) (Fig 4), while differences in IPF patients remained not significant (Fig 5).

### Multivariable analysis of survival

Factors predicting survival are depicted in Table 3. Interestingly, the presence of late native lung problems predicted survival in emphysema patients (OR: 2.55; p = 0.07).

### Discussion

Lung transplantation is the unique treatment that has demonstrated to improve quality of life and survival in patients with end-stage lung parenchymal and vascular diseases, in whom all medical therapy has failed [1].

This surgical therapy differs from the one in other solid organ transplants, as it offers the transplant team the possibility of selecting different transplant procedures: left SLT, right SLT or BLT.

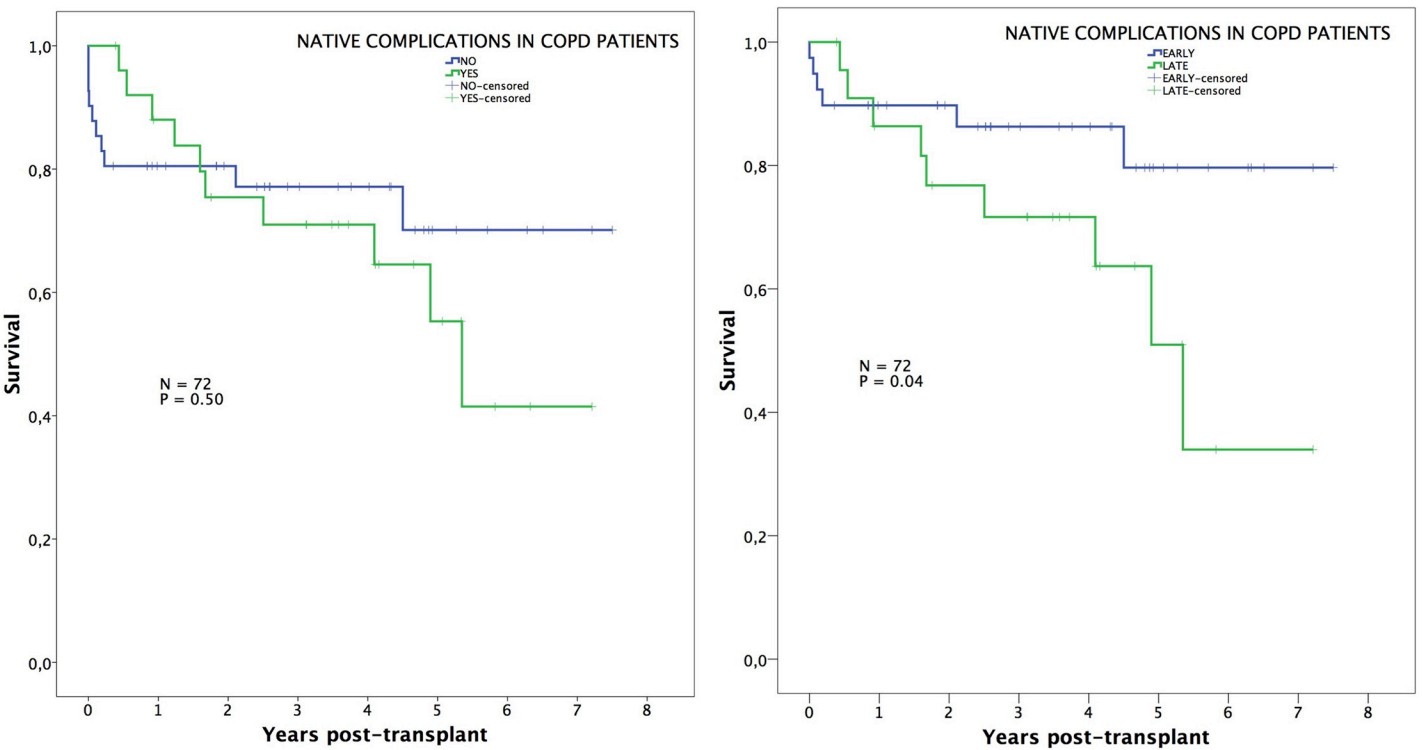

**Fig 3. Survival in COPD patients.** Post-transplant survival of COPD patients comparing those with or without native lung complications (left), and those with early or late native lung complications (right).

Currently, SLT represents around 45% of the lung transplant options [2]. The main advantages of this procedure are a quicker and less aggressive surgery, decreased organ ischemic time, and a more efficient use of the scarce lung donor pool by performing twinning procedures (two recipients transplanted from the same donor), which decreases the mortality rate while on the waiting list [11]. On the other hand, a poorer survival among patients undergoing a SLT has been reported [2, 6, 7].

On the contrary, bilateral lung transplants have demonstrated a better long-term survival due to a greater postoperative pulmonary reserve and to a decreased incidence of CLAD [2–4].

In general, almost all investigations aimed at comparing survival between unilateral and bilateral procedures have been focused on the lung grafts. However, little is known about the role of the native diseased lung as the cause of a worst survival after SLT.

In the present study, we were able to demonstrate that native lung complications adversely affected post-transplant survival, especially in COPD patients, but did not have an impact on 30-day mortality.

Pneumonia, pneumothorax, native lung hyperinflation and lung cancer are the most common complications observed in the native lung after SLT [12–14]. In one of the largest series published to date, these complications occurred in about 14% of all patients after SLT, with a significant reduction in post-transplant survival [14]. This high incidence of native lung complications questions the role of SLT in patients with COPD and IPF and has resulted in recent years in a worldwide shift from SLT to BLT, with better survival results [2].

In the present series, after analysing 161 patients, we observed complications arising in the native lung in 30% of the cases, with a higher incidence in the long-term period. Using smaller

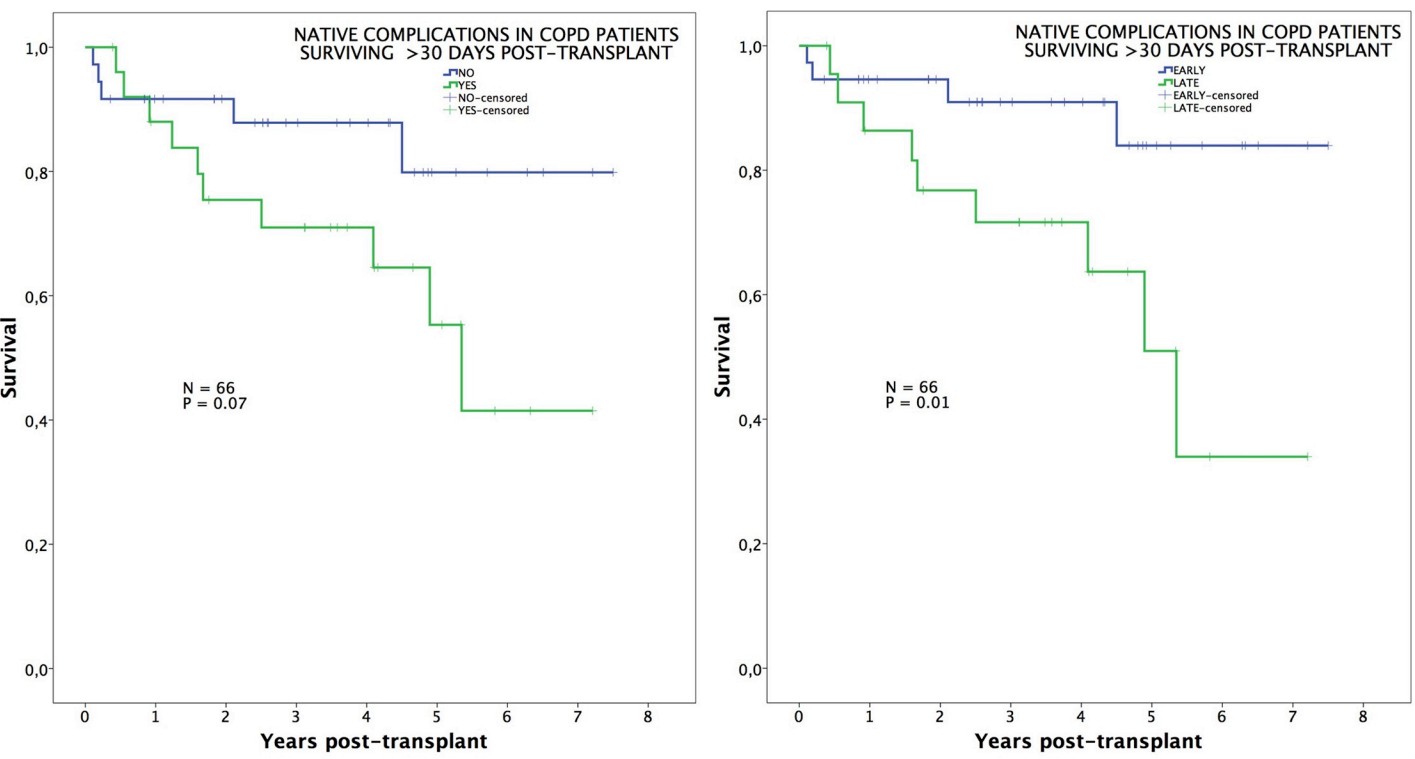

**Fig 4. Survival in COPD patients conditional to survive 30 days.** Post-transplant survival, conditional to survive 30 days, of COPD patients comparing those with or without native lung complications (left), and those with early or late native lung complications (right).

lung transplant populations, other authors have reported an incidence of native lung complications ranging from 14% to 50% [12–15].

Early native lung complications were observed in 9% of the cases. They were mainly atelectasis, pneumonia, pleural effusion, air leaks after LVRS, and pneumothorax, but none of them were itself the cause of 30-day mortality. These complications were strongly associated to longer post-transplant ventilation, ICU, and hospital stay. However, it is noteworthy that early mortality after SLT was not related to problems in the native lung, neither in COPD or IPF patients. So it seems clear that this early mortality after lung transplantation is related to problems in the transplanted lung (graft dysfunction, pneumonia, etc.), or to cardiac adverse events.

Within 30 days post-transplant, atelectasis in the native lung requiring bronchoscopy, was observed in 2% of the cases, and pneumonia in 1% of the cases. Both of these early complications are commonly related to mortality in patients undergoing general thoracic procedures [16]. In line with this observation, Venuta et al. [12] reported one case of atelectasis and one case of pneumonia among their 35 lung transplant recipients. None of these complications were the cause of death in their investigation, as did not in the present study. On the contrary, in our series, the rate of late pneumonia in the native lung leading to death is remarkably high, especially in fibrotic patients.

Three COPD patients (2%) underwent a lung volume reduction surgery in the native lung after completion of the lung transplant. All of them presented prolonged air leaks that added morbidity to the postoperative course, but none of these patients died from this complication and were successfully treated conservatively by maintaining an adequate drainage of the pleural space. Even though BLT is the better option to avoid this complication in the native lung,

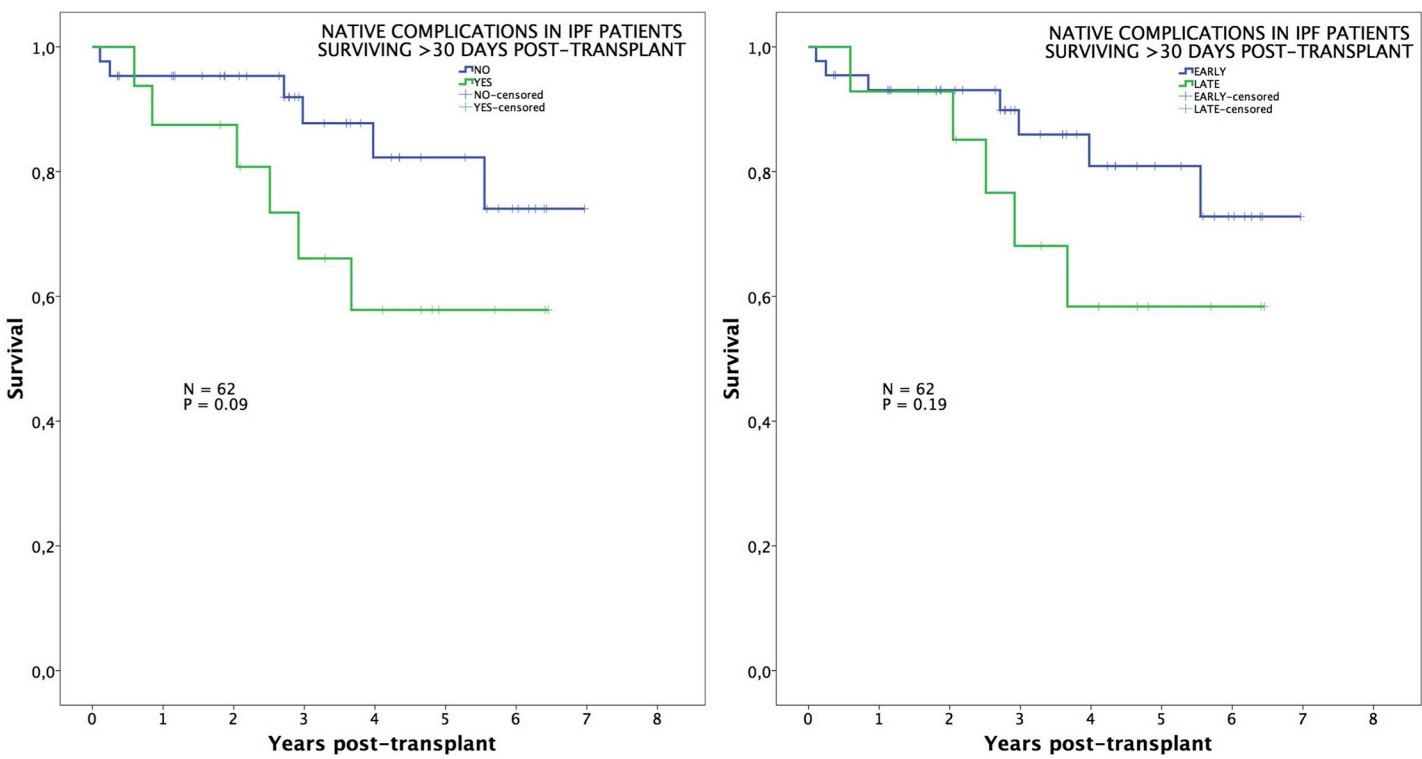

**Fig 5. Survival in IPF patients conditional to survive 30 days.** Post-transplant survival, conditional to survive 30 days, of IPF patients comparing those with or without native lung complications (left), and those with early or late native lung complications (right).

some authors postulated that LVRS after a SLT is an effective treatment strategy with an acceptable surgical risk, but patient selection remains of paramount importance [17].

Postoperative pneumothorax in the native lung was present in 2.5% of our series, similar to that reported by Venuta et al. [12], who treated successfully this complication by VATS. Our patients were successfully treated with pleural drainage in the majority of cases. Only 2 patients required VATS due to persistent air leaks. We strongly recommend indicating early surgery to solve this problem, considering that the subjacent native lung is a diseased organ and the air leak will unlikely resolve spontaneously.

**Table 3. Factors predictive of survival in the overall group, stratified by indication of lung transplantation.**

|  | HR | *95% CI* | p |
|---|---|---|---|
| **OVERALL** |  |  |  |
| CPB/ECMO | 8.20 | 6.52–9.88 | <0.001 |
| Length of postop. ventilation (min.) | 1.99 | 1.79–2.11 | <0.001 |
| Hospital stay (days) | 1.02 | 1.01–1.03 | 0.008 |
| Preop. airway colonizations (no/yes) | 2.26 | 2.12–2.47 | 0.048 |
| **COPD** |  |  |  |
| Native lung late complications (no/yes) | 2.55 | 2.31–2.67 | 0.007 |
| **IPF** |  |  |  |
| Length of postop. ventilation (min.) | 1.99 | 1.82–2.07 | 0.035 |

CPB: cardiopulmonary bypass; ECMO: extracorporeal membrane oxygenation; HR: hazard ratio.

In the long-term, we observed a higher rate of native lung complications reaching up to 25% of our patients. When analysing long-term survival, late native lung complications were strongly associated with a worst survival among COPD patients. On the contrary, IPF patients did not exhibit these differences in survival. Main complications in the non-transplanted lung were hyperinflation and lung cancer.

Hyperinflation was present in 15 COPD patients (9%), leading to a worsening of clinical and functional status. All of these patients presented preoperative bullous emphysema in the preoperative chest CT scan, which support the idea that this technique may be the most useful tool to decide the transplant procedure in order to prevent this complication. Other series have reported lower native hyperinflation rates ranging from 5% to 8% [13, 15], possibly due to the variability in the use of clinical, functional and radiological parameters to describe this complication, and because of the lack of objective parameters to assess hyperinflation and distinguish it from CLAD in the transplanted lung.

In the present series, lung cancer was the most devastating native lung complication, arising in 11 COPD patients (7%). It also impaired long-term survival, with a significant decline of survival in COPD patients presenting lung carcinoma of the native lung. In a previous analysis of the first 340 transplanted patients at our Institution, we identified 9 (2.6%) patients developing lung cancer after lung transplantation, with an interval from transplantation to lung cancer diagnosis of 53.3 ± 12 months and mean survival after cancer diagnosis of 49.3 ± 6.3 months [18].

In one of the first series on lung cancer after lung transplantation including patients from seven U.S. centres, Collins et al. reported an incidence of 2.5% of lung cancer in the native lung after SLT [19]. The group of Leuven recently reported an incidence of developing lung cancer of 9.8% in COPD or IPF patients undergoing SLT. At diagnosis, four patients had local disease (cT1N0M0 and cT2N0M0), whereas all others had loco-regionally advanced or metastatic disease. Five patients were surgically treated and all other patients were treated with chemotherapy with or without radiotherapy [20]. In our series, 5 lung cancer patients were in stages I/II and underwent surgical resection, whereas the remaining 6 cases underwent chemo/radiotherapy.

Other authors have also described lung cancer in the native lung after SLT with an incidence ranging from 0.4% to 8.9% [21–24]. The majority of patients presented with an advanced disease not amenable to surgical treatment. The prognosis of patients treated with chemotherapy and/or radiotherapy alone was poor, with an aggressive and frequently fatal course. Only 25% of patients survived despite treatment, the majority of these patients had early stage lung cancer and underwent curative resection.

These findings suggest that careful surveillance of any change in the native lung after SLT is of paramount importance for early detection of lung cancer.

In the present series, both early and late complications were statistically associated to the presence of bullae and severe air trapping in the COPD group, and to bronchiectasis and preoperative colonizations among the whole study group. These preoperative findings must be carefully assessed by the multidisciplinary team when indicating the type of lung transplant for each candidate.

The present study has several limitations. First, the weaknesses and biases inherent to the retrospective nature of the study. Second, no donor factors were analysed, although the donor selection and retrieval procedure was performed homogenously throughout the period of study, some donor-related factors could have had an influence on the analysis, but not to the degree to invalidate the main results. Third, bilateral procedures were excluded because they were out of the focus of the analysis (the native lung), and therefore, comparisons between bilateral and unilateral procedures were not performed. Finally, minor complications

following the transplant procedures could have been not reported in the clinical charts and therefore not taken into account for the analysis.

## Conclusions

The present study demonstrates that the native lung is a source of morbidity in the short and long term for IPF patients, and appears to be associated to a reduction of long-term survival in COPD patients. The native lung behaviour after SLT should be taken into consideration when choosing the transplant procedure, especially in COPD patients.

## Supporting information

**S1 File. Complete dataset from which the analysis was performed.**
(SAV)

**S2 File. Model of informed consent for organ donation in Spain.**
(PDF)

## Author Contributions

**Conceptualization:** Francisco Javier Gonzalez, Antonio Alvarez.

**Data curation:** Francisco Javier Gonzalez, Enriqueta Alvarez, Paula Moreno, David Poveda, Eloisa Ruiz, Alba Maria Fernandez.

**Formal analysis:** Enriqueta Alvarez, Paula Moreno, Eloisa Ruiz, Antonio Alvarez.

**Investigation:** Francisco Javier Gonzalez, Paula Moreno.

**Methodology:** Antonio Alvarez.

**Supervision:** Angel Salvatierra, Antonio Alvarez.

**Validation:** Antonio Alvarez.

**Writing – original draft:** Francisco Javier Gonzalez, Enriqueta Alvarez, Paula Moreno.

**Writing – review & editing:** Antonio Alvarez.

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
