## [Decision Letter · Decision Letter 0]

3 Feb 2021

PONE-D-21-01459

THE INFLUENCE OF THE NATIVE LUNG ON EARLY OUTCOMES AND SURVIVAL AFTER SINGLE LUNG TRANSPLANTATION

PLOS ONE

Dear Dr. Alvarez,

Thank you for submitting your manuscript to PLOS ONE. After careful consideration, we feel that it has merit but does not fully meet PLOS ONE’s publication criteria as it currently stands. Therefore, we invite you to submit a revised version of the manuscript that addresses the points raised during the review process.

Please address the issues and revise accordingly.

We look forward to receiving your revised manuscript.

Kind regards,

Academic Editor

PLOS ONE

Journal Requirements:

2) We note that your study involved tissue/organ transplantation. Please provide the following information regarding tissue/organ donors for transplantation cases analyzed in your study.

3) Please provide the source(s) of the transplanted tissue/organs used in the study, including the institution name and a non-identifying description of the donor(s).

4) Please state in your response letter and ethics statement whether the transplant cases for this study involved any vulnerable populations; for example, tissue/organs from prisoners, subjects with reduced mental capacity due to illness or age, or minors.

- If a vulnerable population was used, please describe the population, justify the decision to use tissue/organ donations from this group, and clearly describe what measures were taken in the informed consent procedure to assure protection of the vulnerable group and avoid coercion.

- If a vulnerable population was not used, please state in your ethics statement, “None of the transplant donors was from a vulnerable population and all donors or next of kin provided written informed consent that was freely given.

5)  In the Methods, please provide detailed information about the procedure by which informed consent was obtained from organ/tissue donors or their next of kin. In addition, please provide a blank example of the form used to obtain consent from donors, and an English translation if the original is in a different language.

6) Please indicate whether the donors were previously registered as organ donors. If tissues/organs were obtained from deceased donors or cadavers, please provide details as to the donors’ cause(s) of death.

7) Please provide the participant recruitment dates and the period during which transplant procedures were done (as month and year).

8) Please discuss whether medical costs were covered or other cash payments were provided to the family of the donor. If so, please specify the value of this support (in local currency and equivalent to U.S. dollars)."

9) In the ethics statement in the manuscript and in the online submission form, please provide additional information about the patient records/samples used in your retrospective study, including: a) whether all data were fully anonymized before you accessed them; b) the date range (month and year) during which patients' medical records/samples were accessed; c) the source of the medical records/samples analyzed in this work (e.g. hospital, institution or medical center name).

10) In the Ethics Statement on the online submission form and the manuscript Methods , please clarify the context in which consent was obtained, and specify whether patients provided:

    a) Consent to use their medical records/samples used in research

    b) Consent to undergo the procedure

    c) Consent to take part in the study reported in this manuscript.

If the ethics committee waived the need for additional informed consent, please state this.

Thank you for your attention to these requests.

11) We note that you have indicated that data from this study are available upon request. PLOS only allows data to be available upon request if there are legal or ethical restrictions on sharing data publicly. For information on unacceptable data access restrictions, please see http://journals.plos.org/plosone/s/data-availability#loc-unacceptable-data-access-restrictions.

12) Please amend your manuscript to include your abstract after the title page.

Reviewers' comments:

Reviewer's Responses to Questions

**Comments to the Author**

1. Is the manuscript technically sound, and do the data support the conclusions?

Reviewer #1: Yes

Reviewer #2: Partly

Reviewer #3: Partly

2. Has the statistical analysis been performed appropriately and rigorously? 

Reviewer #1: Yes

Reviewer #2: I Don't Know

Reviewer #3: No

3. Have the authors made all data underlying the findings in their manuscript fully available?

Reviewer #1: No

Reviewer #2: Yes

Reviewer #3: Yes

4. Is the manuscript presented in an intelligible fashion and written in standard English?

Reviewer #1: Yes

Reviewer #2: Yes

Reviewer #3: No

5. Review Comments to the Author

Reviewer #1: The manuscript entitled "THE INFLUENCE OF THE NATIVE LUNG ON EARLY OUTCOMES AND SURVIVAL

AFTER SINGLE LUNG TRANSPLANTATION" is well written and a potential topic of interest in the field of lung transplant. The research finding is not novel and there is no direct evidence how the native lung is a source of morbidity. As the author mentioned that this topic is a very controversial, However, due to the correct, appropriately and rigorous data analysis I will recommend this paper for acceptance, and the controversies related to this topic i will leave on the readers.

Reviewer #2: Most of LT are performed as the last therapeutic option in end stage degenerative lung disease as COPD-EMPHYSEMA and fibrotic lung disease , mainly IPF. The last decade was marked by a clear shift to BLT vs SLT due to surgical, prognostic (mortality/survival) causes. Although world statistics still present almost half of patients receiving SLT the BLT trend mainly in very active transplant centers is clear. Focusing on lung grafts as a source of lower survival in SLT is limited as the authors noticed. The native lung was analyzed by various groups before as to its impact on survival

The authors add a local retrospective study to the existing data and looked on the impact of the native lung status on survival following SLT

There ar 2 main flaws in the methodology as far as I understand:

1. one may not decide the impact on survival on patients following SLT using any parameter without comparing to BLT. As mentioned by the authors this lack of comparison is a limitation, in my opinion a fundamental one. You may noy recommend BLT for COPD patients instead of SLT without comparing relevant data from BOTH options

2. I am confused by comparing the survival and mortality data. All patients without native lung complications presented higher 30 days mortality than those with (even when divided between COPD and IPF although not significant).On the othe hand survival was worse in COPD patients with native lung complications (both early and late) but not in IPF patients. There is no clear explanation to this aparent discrepance both in text and figure legends.

Reviewer #3: The topic is of interest for the lung transplant community.

English language is the first main limitation of the manuscript. Many paragraphs are very hard to understand.

The design of the study is somehow confusing and should be clearly presented upfront. The reader is expecting to understand the object of your analysis, the outcome measures, and the methodology of your analysis.

The statistical methodology is also confusing and should be reviewed both for the descriptives and the regression analysis.

I also suggest to redraft your discussion and try to define the key results of the study. The discussion is very long and the take-home messages are not clearly presented. Please shorten your text and make it simple.

Overall, I think that you should consider revising the structure and presentation of your study. The topic is interesting and deserves the work and time of a revision.

6. PLOS authors have the option to publish the peer review history of their article (what does this mean?). If published, this will include your full peer review and any attached files.

Reviewer #1: No

Reviewer #2: No

Reviewer #3: No

---

## [Author Response · Author response to Decision Letter 0]

3 Mar 2021

Dear Editor

We wish to thank you and the reviewers for your consideration and comments regarding our manuscript entitled:

THE INFLUENCE OF THE NATIVE LUNG ON EARLY OUTCOMES AND SURVIVAL AFTER SINGLE LUNG TRANSPLANTATION

(PONE-D-21-01459)

Some concerns have been raised in relation to the use of donors in the present study:

1. Regarding consent for donation: Presumed consent was introduced in Spain by law in 1979. The law establishes that absence of explicit refusal automatically makes the patient a potential donor, but requires that a patient’s possible refusal to donate should be sought by checking their belongings and consulting proxy decision makers. Since most patients have not registered as donors and do not carry donor cards, Spanish transplant coordinators usually have to establish the patient’s wishes through discussion with the family. In practice, organ procurement is not undertaken if the family refuses the donation. Therefore, even though we have a presumed consent policy, we do not apply it in practice. We always approach the relatives, explaining to them the patient’s health conditions and we try to find out whether the individual wanted to be an organ donor or not. If the relatives oppose to deceased organ donation, we do not go on with it.

2. Regarding reimbursements to donor relatives: In some parts of the USA, financial incentives are used as a strategy to increase organ donation— eg, reimbursement for funeral expenses to the family of the deceased individual—despite concerns that excessive sums can constitute a form of unethical inducement. The Spanish Real Decreto 2070–1999 forbids any person from obtaining any kind of financial compensation for human organs and frames organ donation as a voluntary and altruistic act. Therefore, obtaining reimbursements or financial compensations for organ donation in Spain is illegal (Rodriguez-Arias D, Wright L, Paredes D. Success factors and ethical challenges of the Spanish Model of organ donation. Lancet 2010; 376: 1109–12). In addition, the spanish activity on transplantation and donor procurement, follows strictly the regulations approved by the European Parliament: Directive of the European Parliament and of the Council on standards of quality and safety of human organs intended for transplantation: http://www.europarl.europa.eu/sides/getDoc.do?type=TA&reference=P7-TA-2010-0181&format=XML&language=EN Date: May 19, 2010 (accessed Feb 6, 2021).

3. The source of the transplanted organs used in the study, including the donor Institution name (variable: “donor_centre”), city (variable “donor_city”), donor cause of death (variable “donor_death”) and a non-identifying description of the donor including, age, gender, optimal vs. suboptimal, and oxygenation index, are included in the data set submitted as Supporting Information File. This file, not only contains the donor data, but also the complete data set from which the analysis was performed including recruitment dates and the period during which transplant procedures were done (as month and year).

4. None of the transplant donors was from a vulnerable population and all donors or next of kin provided written informed consent that was freely given. This sentence has been included in the Ethics Statement within the text.

In the revised submission, please find a rebuttal letter answering the reviewers’ comments and a detailed explanation of changes in the text.

We have also changed our Data Availability Statement by submitting the complete anonymized data set to replicate our study findings (Supporting Information File).

Thank you for you attention to our manuscript.

Best regards

Antonio Alvarez

RESPONSE TO REVIEWERS

We would like to thank the reviewers of PLOS ONE for taking the time to review our manuscript entitled THE INFLUENCE OF THE NATIVE LUNG ON EARLY OUTCOMES AND SURVIVAL AFTER SINGLE LUNG TRANSPLANTATION. We appreciate the valuable and detailed comments provided by the reviewers. 

We have made some corrections and clarifications in the manuscript after going over the reviewer’s comments.

Reviewer #1: The manuscript entitled "THE INFLUENCE OF THE NATIVE LUNG ON EARLY OUTCOMES AND SURVIVALAFTER SINGLE LUNG TRANSPLANTATION" is well written and a potential topic of interest in the field of lung transplant. The research finding is not novel and there is no direct evidence how the native lung is a source of morbidity. As the author mentioned that this topic is a very controversial, however, due to the correct, appropriately and rigorous data analysis I will recommend this paper for acceptance, and the controversies related to this topic I will leave on the readers.

RESPONSE: We thank the reviewer for his/her comments to our manuscript.

Reviewer #2: Most of LT are performed as the last therapeutic option in end stage degenerative lung disease as COPD-EMPHYSEMA and fibrotic lung disease, mainly IPF. The last decade was marked by a clear shift to BLT vs. SLT due to surgical, prognostic (mortality/survival) causes. Although world statistics still present almost half of patients receiving SLT the BLT trend mainly in very active transplant centers is clear. Focusing on lung grafts as a source of lower survival in SLT is limited as the authors noticed. The native lung was analyzed by various groups before as to its impact on survival

The authors add a local retrospective study to the existing data and looked on the impact of the native lung status on survival following SLT. There are 2 main flaws in the methodology as far as I understand:

1. One may not decide the impact on survival on patients following SLT using any parameter without comparing to BLT. As mentioned by the authors this lack of comparison is a limitation, in my opinion a fundamental one. You may not recommend BLT for COPD patients instead of SLT without comparing relevant data from BOTH options.

RESPONSE: We thank the comments of the reviewer. It is clear that we cannot recommend a bilateral procedure over a unilateral lung transplant on the basis of the analysis or single lung transplants alone. We agree with the reviewer that, for these purposes, the cohort of BLT should have been included in the study. However, the analysis of bilateral procedures was out of the focus of the present analysis. Our purpose has to know what happened with the native lung, irrespective of the behaviour of the lung graft, and whether the native complications could impact on early outcomes and survival. For this reason, according to the suggestions of the reviewer, we have eliminated recommendations of BLT for COPD instead of SLT because this statement is not supported by the data reported in our study.

2. I am confused by comparing the survival and mortality data. All patients without native lung complications presented higher 30 days mortality than those with (even when divided between COPD and IPF although not significant). On the other hand survival was worse in COPD patients with native lung complications (both early and late) but not in IPF patients. There is no clear explanation to this aparent discrepancy both in text and figure legends.

RESPONSE: We thank the reviewer comments. This is one of the major observations of our study: the fact that the native lung is not responsible of major complications leading to death in the early post-transplant period (30-days). Our analysis confirms that 30-day mortality after SLT occurs as a consequence of complications related to the lung graft (not the native), in addition to other well-known causes of early death (cardiac failure, infectious complications). On the contrary, long-term survival was adversely affected by complications in the native lungs, especially in COPD patients with lung cancer arising in the native lung. As shown in figure 1, only one case (IPF group) (2%) died due to native complications (native lung pneumonia), as opposed to 16 (15%) of early deaths that were unrelated to native lung problems. As only one case died in the early post-transplant period as a consequence of a native complication, differences of survival, especially for COPD, are related with late complications (rather than early complications), especially lung neoplasms in the native. This is more clearly demonstrated when analysing survival in the cohort of patients surviving 30 days post-transplant.

Reviewer #3: The topic is of interest for the lung transplant community.

English language is the first main limitation of the manuscript. Many paragraphs are very hard to understand.

RESPONSE: Thank you for your comments. An extensive review of English language has been done. English grammar, expressions, and syntax corrections have been made throughout the text. Some paragraphs have been shortened to make them more understandable.

The design of the study is somehow confusing and should be clearly presented upfront. The reader is expecting to understand the object of your analysis, the outcome measures, and the methodology of your analysis.

RESPONSE: Thank you for your comments. This is an observational analytic retrospective case-control study to determine the influence of native lung complications on early outcomes and survival after SLT for IPF or COPD. The primary end-points are 30-day mortality and survival. This clarification has been included in methods section (lines 86-96).

The statistical methodology is also confusing and should be reviewed both for the descriptives and the regression analysis.

RESPONSE: Thank you for your valuable comments. The description of the statistical methods was reviewed by our statistician without observing major flaws in the description. Nevertheless, changes have been included to better describe the statistics, according to the suggestions of the reviewer (lines 137-141). Also, we have included information regarding the availability of the data set used for the present analysis (lines 176-180). 

I also suggest to redraft your discussion and try to define the key results of the study. The discussion is very long and the take-home messages are not clearly presented. Please shorten your text and make it simple.

RESPONSE: Thank you for you comments. We agree with the reviewer that the Discussion is too long. We have made extensive changes throughout the text to make it clearer. We believe that the main message is more clearly presented.

Overall, I think that you should consider revising the structure and presentation of your study. The topic is interesting and deserves the work and time of a revision.

RESPONSE: We wish to thank the reviewer for taking the time to review our manuscript.

---

## [Decision Letter · Decision Letter 1]

25 Mar 2021

THE INFLUENCE OF THE NATIVE LUNG ON EARLY OUTCOMES AND SURVIVAL AFTER SINGLE LUNG TRANSPLANTATION

PONE-D-21-01459R1

Dear Dr. Alvarez,

We’re pleased to inform you that your manuscript has been judged scientifically suitable for publication and will be formally accepted for publication once it meets all outstanding technical requirements.

Kind regards,

Academic Editor

PLOS ONE

Additional Editor Comments (optional):

Reviewers' comments:

Reviewer's Responses to Questions

**Comments to the Author**

1. If the authors have adequately addressed your comments raised in a previous round of review and you feel that this manuscript is now acceptable for publication, you may indicate that here to bypass the “Comments to the Author” section, enter your conflict of interest statement in the “Confidential to Editor” section, and submit your "Accept" recommendation.

Reviewer #1: All comments have been addressed

Reviewer #4: All comments have been addressed

2. Is the manuscript technically sound, and do the data support the conclusions?

Reviewer #1: Yes

Reviewer #4: Yes

3. Has the statistical analysis been performed appropriately and rigorously? 

Reviewer #1: Yes

Reviewer #4: Yes

4. Have the authors made all data underlying the findings in their manuscript fully available?

Reviewer #1: No

Reviewer #4: Yes

5. Is the manuscript presented in an intelligible fashion and written in standard English?

Reviewer #1: Yes

Reviewer #4: Yes

6. Review Comments to the Author

Reviewer #1: The author properly addressed the reviewers recommendation point by point, and I can recommend to accept the manuscript in its current form.

Reviewer #4: Dear Authors,

In the updated version of your manuscript investigating on short- and long-term mortality after lung transplantation in COPD and IPF patients, you exhaustively addressed all the reviewr comments.

Regards

7. PLOS authors have the option to publish the peer review history of their article (what does this mean?). If published, this will include your full peer review and any attached files.

Reviewer #1: No

Reviewer #4: No

---

## [Editor Report · Acceptance letter]

29 Mar 2021

PONE-D-21-01459R1 

THE INFLUENCE OF THE NATIVE LUNG ON EARLY OUTCOMES AND SURVIVAL AFTER SINGLE LUNG TRANSPLANTATION 

Dear Dr. Alvarez:

I'm pleased to inform you that your manuscript has been deemed suitable for publication in PLOS ONE. Congratulations! Your manuscript is now with our production department. 

Kind regards, 

on behalf of

Dr. Robert Jeenchen Chen 

Academic Editor

PLOS ONE